# Radiomics Combined with Multiple Machine Learning Algorithms in Differentiating Pancreatic Ductal Adenocarcinoma from Pancreatic Neuroendocrine Tumor: More Hands Produce a Stronger Flame

**DOI:** 10.3390/jcm11226789

**Published:** 2022-11-16

**Authors:** Tao Zhang, Yu Xiang, Hang Wang, Hong Yun, Yichun Liu, Xing Wang, Hao Zhang

**Affiliations:** 1State Key Laboratory of Biotherapy, Sichuan University, Chengdu 610041, China; 2Department of Biotherapy, West China Hospital, Sichuan University, Chengdu 610041, China; 3West China School of Medicine, Sichuan University, Chengdu 610041, China; 4Sichuan Center for Disease Control and Prevention, Chengdu 610041, China; 5Department of Pancreatic Surgery, West China Hospital, Sichuan University, Chengdu 610041, China

**Keywords:** radiomics, machine learning, artificial intelligence, pancreatic ductal adenocarcinoma, pancreatic neuroendocrine tumor, diagnostic model

## Abstract

The aim of this study was to assess the diagnostic ability of radiomics combined with multiple machine learning algorithms to differentiate pancreatic ductal adenocarcinoma (PDAC) from pancreatic neuroendocrine tumor (pNET). This retrospective study included a total of 238 patients diagnosed with PDAC or pNET. Using specialized software, radiologists manually mapped regions of interest (ROIs) from computed tomography images and automatically extracted radiomics features. A total of 45 discriminative models were built by five selection algorithms and nine classification algorithms. The performances of the discriminative models were assessed by sensitivity, specificity and the area under receiver operating characteristic curve (AUC) in the training and validation datasets. Using the combination of Gradient Boosting Decision Tree (GBDT) as the selection algorithm and Random Forest (RF) as the classification algorithm, the optimal diagnostic ability with the highest AUC was presented in the training and validation datasets. The sensitivity, specificity and AUC of the model were 0.804, 0.973 and 0.971 in the training dataset and 0.742, 0.934 and 0.930 in the validation dataset, respectively. The combination of radiomics and multiple machine learning algorithms showed the potential ability to discriminate PDAC from pNET. We suggest that multi-algorithm modeling should be considered for similar studies in the future rather than using a single algorithm empirically.

## 1. Introduction

Pancreatic cancer is one of the most common malignancies with an increasing incidence and is the seventh leading cause of cancer death due to poor prognosis [1,2,3]. Based on its origins and pathology, pancreatic cancer has been classified into several subtypes, including pancreatic ductal adenocarcinoma (PDAC) and pancreatic neuroendocrine tumor (pNET). Among them, PDAC, accounting for 90% of all primary pancreatic malignancies, shows aggressive behavior and poor prognosis, which leads to a five-year survival rate of less than 5% [4,5]. It can only be cured by surgical resection, which is available merely to a small number of patients whose tumors can be surgically removed before they progress to the advanced stage [2]. Unfortunately, most patients tend to be diagnosed with metastasis, in which case chemotherapy and radiotherapy do not significantly improve survival rates [6]. Even immunotherapy, which has shown success in other tumors in recent years, is ineffective in treating PDAC [7]. In contrast, pNET is not as common as PDAC and carries a significantly better prognosis with a five-year survival rate of 51.3% [8]. Pancreatic neuroendocrine tumors are classified into functioning or non-functioning according to clinical symptoms. The tumor cells in functioning pNET can secrete a variety of hormones, such as glucagon, insulin and gastrin, which can lead to symptoms of hormone hypersecretion [9]. Non-functioning pNET does not produce excess hormones but has a wide range of clinical behaviors [10]. The treatment strategy for pNET includes surgical resection and chemotherapy, while hormone therapy is controversial. Contrary to PDAC, pNET patients can benefit from early treatment and long-term survival can be expected even with metastases [11]. Therefore, the importance of an early diagnosis of PDAC and pNET cannot be overstated due to their different prognosis and therapeutic strategies.

Magnetic resonance imaging (MRI) and computed tomography (CT) are widely used in detecting abdominal lesions and lymph node metastases. However, MRI has some limitations compared with CT, such as it being time-consuming and costly, and patients experience noise and isolation during the scan. CT is considered as an economical radiological exam to detect pancreatic tumors [12]. Nevertheless, the CT characteristics of PDAC and pNET are quite similar, which makes it easy to lead to misdiagnosis by visual assessment [13,14]. Consequently, it is necessary to develop more feasible methods to assist in diagnosis and improve accuracy. The emergence of radiomics has shown promising prospects in the domain of radiological evaluation. Radiomics refers to the high-throughput extraction of quantitative image features from medical imaging followed by data analysis to support clinical decision-making [15,16,17]. At the same time, new methods for analyzing mineable radiomics feature are required. Machine learning is a major branch of artificial intelligence and is defined as the ability of a machine to learn and predict future events and outcomes based on large datasets [18]. The combination of radiomics and machine learning has presented promising performance in previous research, including predicting histological subtypes of lung cancer, the grade of meningioma and survival outcomes in non-small-cell lung cancer [19,20,21]. Moreover, it has also been prominent in the diagnosis of pancreatic lesions such as pseudocysts, serous cystadenomas, autoimmune pancreatitis and PDAC [22,23,24]. Previous studies have explored the ability to differentiate PDAC from pNET by purely textural features based on CT or MRI in relatively small sample sizes [25,26].

Although radiomics and machine learning have been widely used in disease diagnosis, the application of radiomics and multiple machine learning algorithms combined in discriminating PDAC and pNET has not been reported. The aim of this study was to evaluate the diagnostic ability of radiomics features combined with multiple machine learning algorithms in differentiating PDAC from pNET.

## 2. Materials and Methods

### 2.1. Patient Selection

We retrospectively viewed our database to search for patients diagnosed with PDAC or pNET in West China Hospital with detailed medical records from August 2013 to May 2019. The eligibility criteria for the inclusion of selected patients were: (1) with complete medical records; (2) with a pathological diagnosis confirmation of PDAC or pNET; (3) with high-quality CT images before surgery. Exclusion criteria were: (1) history of treatment before CT scan, such as chemotherapy and radiotherapy; (2) history of other cancers. A total of 238 patients with PDAC or pNET met the inclusion criteria in the initial selection. Then, we collected the information of these patients, including gender, age, body mass index (BMI), glucose, calcium, procalcitonin, α-fetoprotein (AFP), carcinoembryonic antigen (CEA), glucoprotein antigen 199 (CA19-9), glucoprotein antigen 125 (CA12-5), total bilirubin (TBIL), amylase and lipase. This study was approved by the Ethics Committee of West China Hospital, Sichuan University, and all patients’ informed consents were waived.

### 2.2. Image Acquisition and Texture Feature Extraction

All selected patients underwent CT examinations, the operating parameters of which were reported in previous studies [27]. The procedures of scanning are summarized in the Appendix A. The CT diagnostic criteria for pancreatic cancer are divided into three parts: morphologic evaluation, vascular evaluation and extrapancreatic evaluation [28]. Morphologic evaluation includes appearance (confined masses of the pancreas are less dense than normal pancreas), size, location (most commonly PDAC in the head of the pancreas and neuroendocrine elsewhere), pancreatic duct narrowing/abrupt cut-off with or without upstream dilatation, and biliary tree abrupt cut-off with or without upstream dilatation. Vascular evaluation can be divided into arterial evaluation and venous evaluation. Arterial evaluation includes the evaluation of the celiac axis, superior mesenteric artery (SMA), common hepatic artery (CHA) and arterial variants. Venous evaluation includes the evaluation of the main portal vein (MPV), superior mesenteric vein (SMV), venous collaterals and intravenous thrombosis. Extrapancreatic evaluation includes the evaluation of liver metastases, peritoneal nodes, suspicious lymph nodes in the abdomen and ascites. PDAC is considered if the above criteria for pancreatic cancer are met, otherwise pNET is suspected, but the final diagnosis is based on pathological results. Recognizable CT images with clear boundaries were acquired from the picture archiving and communication systems in the radiology department. Two radiologists with pancreatic expertise extracted the texture features from CT images using the Local Image Features Extraction (LIFEx) software (v3.74, CEA-SHFJ, Orsay, France) [29]. In the guidelines of the software, ROI was manually drawn along the border of tumor issue slice by slice to gain the 3D features (Figure 1). Any differences in ROIs were recorded and discussed. The preferred ROI of each patient was selected by the third senior radiologist and was included in our study. All ROI data from the enhanced CT were labeled as PDAC or pNET according to the pathological results. Finally, a total of 48 radiomics features were automatically exported and saved through the LIFEx software.

### 2.3. Discriminative Model Establishment and Data Analysis

The selection algorithm is considered as an important data preprocessing step in machine learning tasks, due to its promising ability of reducing the dimensionality of the data. Moreover, it contributes to better machine learning models with a higher prediction accuracy and less training time [30]. We employed 5 selection algorithms in our study, which were Distance Correlation (DC), Random Forest (RF), Least Absolute Shrinkage and Selection Operator (LASSO), eXtreme Gradient Boosting (Xgboost) and Gradient Boosting Decision Tree (GBDT). In addition, 9 classification algorithms were adopted on the basis of the selection methods. The 9 classification algorithms were Linear Discriminant Analysis (LDA), Support Vector Machines (SVM), Random Forest (RF), Adaptive Boosting (AdaBoost), K-nearest neighborhood (KNN), Gaussian Naive Bayes (GaussianNB), Logistic Regression (LR), GBDT and Decision Tree (DT). In this way, 45 diagnostic models were built with cross combinations of 5 different feature-selection algorithms and 9 classification algorithms [31]. The included patients were randomly assigned into training and validation datasets at a ratio of 3:1. The models were initially trained in the training dataset and then were validated in the validation dataset. In addition, these two processes were repeated 10-fold to ensure the accuracy and robustness. Sensitivity, specificity and AUC were all calculated in the training and validation datasets. All machine learning processes were programmed in the python programming language (sklearn package). As for clinical data analysis, the chi-square test, independent-sample *t*-test, Mann–Whitney U-test and Pearson’s correlation coefficient were conducted in the statistical package IBM SPSS Statistics software (Version 26.0, IBM, Armonk, NY, USA).

## 3. Results

### 3.1. Patient Characters

A total of 238 patients were included in this study. Among these patients, 156 patients were diagnosed as PDAC and 82 patients were diagnosed as pNET. The gender ratios (male: female) for each subtype were 97:59 and 48: 34, respectively. The mean ages of patients were 59.69 and 53.39 (*p* < 0.001), respectively. The laboratory examination showed increased glucose levels of PDAC patients (6.83 mmol/L vs. 5.45 mmol/L, *p* < 0.001). PDAC patients had higher levels of CA19-9 than pNET patients (380.10 U/mL vs. 137.91 U/mL, *p* < 0.001). The level of TBIL in PDAC patients was significantly higher than that of pNET patients (115.31 umol/L vs. 26.07 umol/L, *p* < 0.001). There were no significant differences between gender, BMI, calcium, procalcitonin, AFP, CEA, CA12-5, amylase and lipase (*p* > 0.05). The clinical characters and laboratory indexes of patients are shown in Table 1.

### 3.2. Radiomics Features

A total of forty-eight features were extracted from six matrixes. The correlation of these features was tested by Pearson’s correlation coefficient. The results suggested that most of the features were independent. Some features were shown to have a positive correlation, such as GLCM_Entropy and HISTO_Entropy (Pearson’s correlation = 0.971), GLZLM_GLNU and GLRLM_GLNU (Pearson’s correlation = 0.923) and GLZLM_LZHGE and GLZLM_LZE (Pearson’s correlation = 0.998). The heat map of correlation among the radiomics features is shown in Figure 2.

### 3.3. Model Assessment

A total of forty-five models were developed with a combination of five selection algorithms and nine classification algorithms. Radiomics features were introduced into the models. All the discriminative models exhibited a feasible diagnostic ability in the validation dataset. The optimal model in this study was established by the selection algorithm (GBDT) and classification algorithm (RF), which presented the best diagnostic performance with the highest AUC in the training dataset as well as in the validation dataset. This model demonstrated sensitivity, specificity and AUC values in the training dataset of 0.804, 0.973 and 0.971, respectively; in the validation dataset they were 0.742, 0.934 and 0.93, respectively. In the training dataset, the positive and negative prediction values were 0.950 and 0.886, respectively; in the validation dataset, the positive and negative prediction values were 0.898 and 0.822, respectively. The sensitivity, specificity and AUC of the 45 models in the training and validation datasets are summarized in Table 2.

In addition, the combination of RF + SVM and GBDT + SVM showed over-fitting, so we finally excluded them from the model performance comparison. To assess the robustness of the discriminative models, the processes of training and validating were repeated 10 times. The receiver operating characteristic curves (ROCs) of the optimal discriminative model in the cross validation of all folds are shown in Figure 3.

## 4. Discussion

The incidence and mortality of pancreatic cancer varies in different regions of the world. Globally, pancreatic cancer is the seventh leading cause of cancer-related death and the fourth leading cause of cancer-related death in western countries [3,32]. The possible reasons for this difference we believe include dietary habits, obesity rates, level of economic development and ability to treat cancer. Despite having regional differences in incidence and mortality, pancreatic cancer remains one of the major challenges in the diagnosis and treatment of cancer worldwide. As a major subtype of pancreatic cancer, PDAC has unparalleled research value.

The accurate diagnosis of PDAC and pNET is important because of their different treatments and prognoses. CT is widely used as a noninvasive examination approach of abdominal tumors. However, accurate diagnosis by naked eye assessment is challenging due to the similar radiological features shared by PDAC and pNET. In this study, we extracted quantitative features from CT images and established diagnosis models, which combined radiomics with multiple machine learning algorithms to differentiate PDAC from pNET. In addition, the diagnostic ability of 45 models was investigated, demonstrating stable and outstanding discriminative performances. It is worth noting that the combination of GBDT and RF was preferred for the statistical analysis and presented an optimal discriminative ability.

In the regular statistical analysis of baseline information, differences between some parameters were significant. Our study found that the CA19-9 level of PDAC patients was significantly higher than that of pNET patients, which was consistent with previous studies reporting a significant difference in CA19-9 between PDAC patients and pNET patients [33]. The results of reduced blood glucose levels in pNET could be explained by hyperinsulinemic hypoglycemia [9]. In addition, our results suggested that the differences in bilirubin levels were significant in PADC patients and pNET patients. Other researchers have also mentioned hyperbilirubinaemia due to cholestasis in PDAC, and elevated TBIL was associated with a poorer prognosis in patients with pancreatic cancer [34,35]. Except for biochemical indexes, clinical parameters were associated with cancer type. Our study found a significant difference in the mean age between the two types of tumors, which reached a consensus with previous studies [36]. The results demonstrated that the mean age of PDAC patients was higher than that of pNET patients, suggesting that older patients are more likely to suffer from malignant tumors. In addition, it is generally accepted that PDAC usually occurs in older patients, with some studies suggesting that the median age of PDAC patients at diagnosis is 71 [2]. Comparatively, the mean age of the PDAC patients in this study was extremely younger (59.69 years). The possible reason for this discrepancy may be the trend towards a younger incidence of PDAC. It may also be that advances in diagnostic tools have led to an increase in early PDAC detection rates. Similar results have been reported in recent studies, where the mean age of PDAC patients was also significantly lower [37,38]. However, this finding needs to be treated with caution, as it is only the conclusion of a single-center retrospective study conducted in China from 2013 to 2019.

The current diagnostic methods of pancreatic cancer are biopsy, biomarkers and imaging tools [39]. Cytological examinations of clinically unresectable tumors are crucial for determining prognosis and treatment. However, the procedure of biopsy is associated with complications such as infection and hemorrhage and has a risk of peritoneal seeding [12,40]. Additionally, the pathological diagnosis between PDAC and pNET is sometimes mistaken because both of them demonstrate nuclear atypia, necrosis and a high mitotic rate [41]. Biomarkers have a limited role in differentiating PDAC from pNET [41]. Serum chromogranin A (CgA) is widely used as a biomarker for pNET, but it presents a sensitivity and specificity of 60% and 75%, respectively [42]. Moreover, the majority of PDAC patients have also been found with elevated CgA levels [41]. In terms of glucose, it varies in different subtypes of pNET. Insulinoma, the most common pNET, presents a low blood glucose level, while glucagonoma is characterized by diabetes mellitus [9]. However, it has been reported that 85% of PDAC patients are diagnosed with hyperglycemia due to dysregulation of the glucose metabolism [43]. Therefore, blood glucose level is not sufficient for predicting subtypes of pancreatic cancer. In regard to radiological evaluation, the diagnosis of PDAC and pNET by conventional CT depends on the differences in radiological features, which are induced by tumor growth. Originating from endocrine cells or the pluripotent duct cells of the pancreas, pNET usually appears as a solid avidly enhancing mass [14]. However, in some cases, ductal obstruction, ductal dilatation and up-stream pancreatic atrophy could be caused by related metabolites of pNET [25]. Thus, these atypical characteristics and insufficient radiological features make accurate diagnosis difficult by visual assessment. Diagnosis may be delayed due to non-specific clinical symptoms and the lack of effective diagnostic methods, leading to larger primary tumors or even metastases at the time of initial diagnosis [44].

Radiomics provides more quantitative information beyond the information provided by naked eye assessment, including lesion shape, volume and texture [45]. The processed data are too numerous for radiologists to perform a complete visual assessment. However, it can be further analyzed by machine learning, which has the potential to deal with complex tasks, thus improving diagnostic accuracy, optimizing clinical workflow and decreasing costs and workload [46]. The combination of radiomics and machine learning provides promising radiological evaluation methods. In previous studies, researchers investigated various algorithms and radiomics features of pancreatic lesions. Some researchers applied a few volumetric CT texture features to distinguish PADC from pNET, reporting that the combination of fifth percentile + skewness generates the highest AUC of 0.887, and the corresponding sensitivity and specificity were 0.9 and 0.8, respectively [47]. However, the limited number of texture parameters could not reflect the heterogeneity of tumors. Other researchers developed three models to differentiate nonfunctional neuroendocrine tumor (NF-pNET) from PDAC, including a model based on radiomics signatures alone, one based on clinical parameters alone and another model that integrated both [25]. It impressed us that not only radiomics features but clinical parameters were integrated into the models. The results showed that the AUC of the integrated model hit 0.884, improving on the discriminative ability based on radiomics features and clinical parameters alone. Although their model only used a single algorithm, it showed great discriminative ability. However, the small sample size of this study reduced the reliability of its conclusions. In the diagnosis of PDAC and pNET, our study included more algorithms and showed better a discriminative ability with a higher AUC. This might provide a noninvasive method to distinguish those subtypes of pancreatic cancer with a high specificity and sensitivity. Compared with similar studies, our sample size was relatively larger, which greatly increased the reliability of our research conclusions. In addition to CT, some studies have built differentiation models of PNET and PDAC based on MRI or PET-CT, and these models also have excellent differentiation abilities [26,48,49]. In addition to the small sample size, most of these studies only adopted a single algorithm and perhaps did not select the most suitable algorithm, which made their conclusions limited.

Most of the studies about machine leaning in assisting diagnosis showed the results of single machine leaning method or different combinations of algorithms. For example, a study conducted to distinguish axillary lymph node status reported that LDA classifier presented the highest AUC of 0.81 in a 2D analysis [50]. Another researcher reported that RF showed the highest AUC of 0.968 in distinguishing benign ovarian tumors from epithelial ovarian cancer [51]. A study about the diagnosis of glioblastoma and metastatic brain tumors noted that two models combining DC and LDA and combining DC and LR achieved the highest AUC of 0.80 in the validating dataset [52].

However, for the best discriminative models obtained in different studies, the respective algorithms used are not exactly the same. There is no best answer to the choice of algorithms for studies related to radiomics combined with machine learning.

Though there are numerous algorithms for researchers to choose from, previous studies have employed only a few classifiers for discrimination. Therefore, we included more algorithms to evaluate their discriminative performance and tried to select the optimal algorithm. In our study, five selection algorithms and nine classification algorithms were adopted to determine the suitable discriminative models. Our research found that the combination of GBDT and RF presented the best performance. GBDT, a widely used algorithm in machine learning, receives promising predictability when coping with numerous factors and complicated relations [53]. Other researchers suggested that GBDT showed the highest performance in predicting colorectal cancer compared with RF and SVM, indicating that implementing a GBDT model is time-saving and cost-effective [54]. So, GBDT was applicable for our study because of its excellent prediction and ability to deal with numerical values. RF, a collection of decision trees, can balance data and be employed in classification tasks [55,56]. In a previous study, researchers adopted RF for assisting with Parkinson’s disease, suggesting that RF achieved better classification compared with LR and SVM [57]. Considering the various choices of algorithms and different types of cancer, more attention needs to be paid in feature-selection algorithms and machine-learning classifiers in future investigations to build more reliable models. Though a finite number of open-source algorithms were employed in our research, the results of our study could provide a reference for algorithm selection when predicting subtypes of cancer with machine learning.

There were several limitations in our study. First, this was a retrospective study and therefore bias was inevitable. Additionally, we established the model and tested the performance in the database of our institution, and an external validation set from another institution was required.

## 5. Conclusions

In conclusion, methods based on enhanced CT radiomics features combined with multiple machine learning algorithms have a promising ability to differentiate PDAC from pNET. In addition, this assistance method is expected to facilitate clinical decision-making and reduce the invasive injury caused by pathological examinations. We also suggest that when similar research is conducted in the future multiple algorithms should be considered in the model rather than empirically using only a single algorithm.

## Figures and Tables

**Figure 1 jcm-11-06789-f001:**
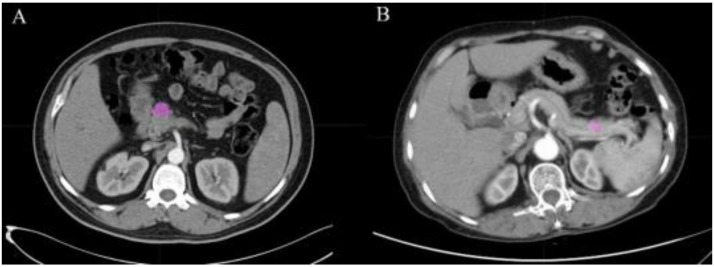
ROI manually drawn from two cases in CT images. (**A**) Example of a 51 year old male with PDAC with lesion in pancreatic head. (**B**) Example of a 60 year old female with pNET with lesion in pancreatic body. The pink areas in the figure are the delineated ROIs.

**Figure 2 jcm-11-06789-f002:**
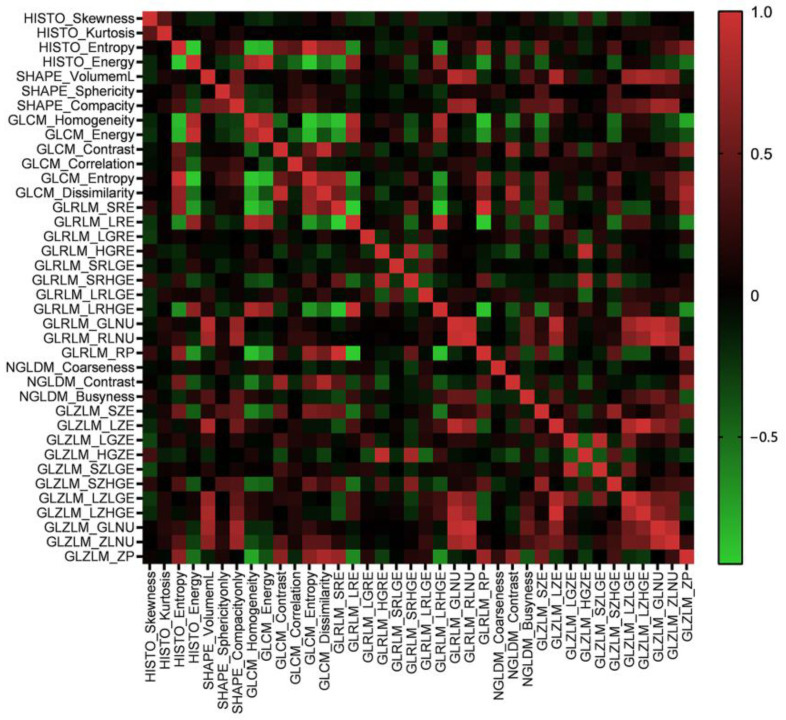
The heat map of Pearson’s correlation coefficients among radiomics features.

**Figure 3 jcm-11-06789-f003:**
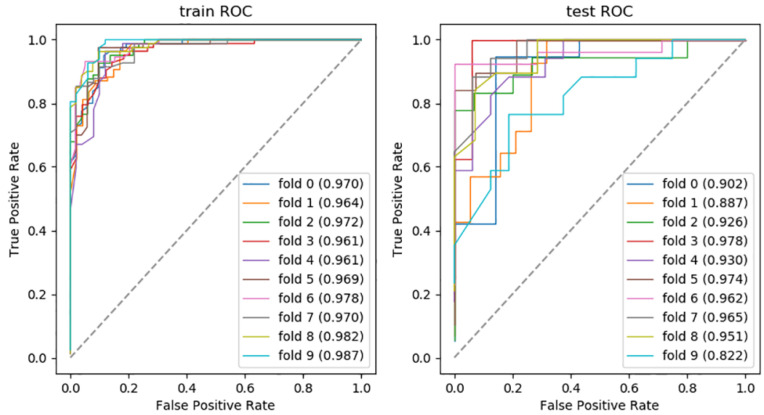
The AUC of optimal discriminative model.

**Table 1 jcm-11-06789-t001:** Characteristics of patients.

Characteristics	PDAC (*n* = 156)	pNET (*n* = 82)	*p* Value
Male gender, *n* (%)	97 (62.2%)	48 (58.5%)	0.307
Mean age (year)	59.69 ± 11.57	53.39 ± 13.01	<0.001 *
BMI (kg/m^2^)	25.26 ± 5.67	21.64 ± 4.25	0.701
Glucose (mmol/L)	6.83 ± 3.47 (*n* = 137)	5.45 ± 3.39 (*n* = 76)	<0.001 *
Procalcitonin (ng/mL)	1.06 ± 2.22 (*n* = 115)	0.92 ± 1.18 (*n* = 58)	0.812
AFP (ng/mL)	3.90 ± 8.58 (*n* = 151)	3.114 ± 1.70 (*n* = 68)	0.434
CEA (U/mL)	6.24 ± 10.01 (*n* = 153)	3.78 ± 6.68 (*n* = 77)	0.092
CA19-9 (U/mL)	380.10 ± 380.60 (*n* = 155)	137.91 ± 282.90 (*n* = 79)	<0.001 *
CA125 (U/mL)	48.96 ± 49.41 (*n* = 134)	20.54 ± 15.04 (*n* = 66)	0.184
TBIL (umol/L)	115.31 ± 144.51 (*n* = 156)	26.07 ± 51.74 (*n* = 82)	<0.001*
Amylase (IU/L)	111.64 ± 107.14 (*n* = 140)	225.13 ± 651.94 (*n* = 70)	0.208
Lipase (IU/L)	183.38 ± 270.75 (*n* = 127)	309.11 ± 1058.07 (*n* = 65)	0.312

* *p* < 0.05; PDAC, pancreatic ductal adenocarcinoma; pNET, pancreatic neuroendocrine tumor; BMI, body mass index; AFP, α-fetoprotein; CEA, carcinoembryonic antigen; CA19-9, glucoprotein antigen 199; CA125, glucoprotein antigen 125; TBIL, total bilirubin.

**Table 2 jcm-11-06789-t002:** Discriminative performance of models.

	Training Dataset			Validation Dataset		
	Sensitivity	Specificity	AUC	Sensitivity	Specificity	AUC
DC + LDA	0.839	0.903	0.953	0.775	0.884	0.938
DC + SVM	0.972	0.563	0.931	0.987	0.528	0.910
DC + RF	0.730	0.963	0.935	0.653	0.929	0.879
DC + AdaBoost	1.000	1.000	1.000	0.753	0.905	0.919
DC + KNN	0.834	0.924	0.961	0.769	0.872	0.884
DC + GaussianNB	0.810	0.851	0.922	0.781	0.796	0.896
DC + LR	0.759	0.903	0.926	0.703	0.863	0.901
DC + GBDT	1.000	1.000	1.000	0.758	0.879	0.918
DC + DT	1.000	1.000	1.000	0.793	0.841	0.817
RF + LDA	0.797	0.95	0.954	0.775	0.929	0.927
RF + SVM	1.00	1.00	-	0.00	1.00	-
RF + RF	0.745	0.974	0.957	0.692	0.931	0.902
RF + AdaBoost	1.000	1.000	1.000	0.759	0.912	0.921
RF + KNN	0.608	0.917	0.886	0.395	0.830	0.707
RF + GaussianNB	0.632	0.956	0.917	0.604	0.921	0.883
RF + LR	0.796	0.897	0.948	0.816	0.878	0.944
RF + GBDT	1.000	1.000	1.000	0.746	0.884	0.927
RF + DT	1.000	1.000	1.000	0.732	0.832	0.782
LASSO + LDA	0.672	0.916	0.878	0.633	0.883	0.811
LASSO + SVM	0.718	0.846	0.814	0.712	0.803	0.784
LASSO + RF	0.709	0.929	0.917	0.626	0.859	0.847
LASSO + AdaBoost	1.000	1.000	1.000	0.660	0.852	0.828
LASSO + KNN	0.659	0.955	0.915	0.601	0.894	0.796
LASSO + GaussianNB	0.328	0.961	0.820	0.357	0.929	0.766
LASSO + LR	0.364	0.986	0.819	0.340	0.963	0.788
LASSO + GBDT	1.000	1.000	1.000	0.677	0.833	0.849
LASSO + DT	1.000	1.000	1.000	0.663	0.842	0.753
Xgboost + LDA	0.808	0.881	0.945	0.817	0.876	0.943
Xgboost + SVM	0.914	0.761	0.939	0.936	0.713	0.951
Xgboost + RF	0.750	0.980	0.960	0.711	0.968	0.925
Xgboost + AdaBoost	1.000	1.000	1.000	0.821	0.916	0.932
Xgboost + KNN	0.804	0.941	0.967	0.777	0.928	0.940
Xgboost + GaussianNB	0.848	0.801	0.918	0.840	0.739	0.896
Xgboost + LR	0.815	0.866	0.938	0.851	0.855	0.948
Xgboost + GBDT	1.000	1.000	1.000	0.758	0.925	0.931
Xgboost + DT	1.000	1.000	1.000	0.783	0.868	0.825
GBDT + LDA	0.861	0.914	0.966	0.827	0.886	0.945
GBDT + SVM	1.000	1.000	-	0.000	1.000	-
GBDT + RF	0.804	0.973	0.971	0.742	0.934	0.930
GBDT + AdaBoost	1.000	1.000	1.000	0.799	0.890	0.929
GBDT + KNN	0.620	0.942	0.881	0.429	0.866	0.730
GBDT + GaussianNB	0.329	0.954	0.907	0.373	0.928	0.884
GBDT + LR	0.809	0.920	0.953	0.782	0.882	0.927
GBDT + GBDT	1.000	1.000	1.000	0.746	0.893	0.927
GBDT + DT	1.000	1.000	1.000	0.735	0.828	0.781

Abbreviations: AUC, area under the receiver operating characteristic curve; DC, Distance Correlation; RF, Random Forest; LASSO, Least Absolute Shrinkage and Selection Operator; Xgboost, eXtreme Gradient Boosting; GBDT, Gradient Boosting Decision Tree; LDA, Linear Discriminant Analysis; SVM, Support Vector Machines; AdaBoost, Adaptive Boosting; KNN, K-nearest neighborhood; GaussianNB, Gaussian Naive Bayes; LR, Logistic Regression; DT, Decision Tree—means that the model is over-fitting, so the results are not displayed.

## Data Availability

The data presented in this study are available on request from the corresponding author.

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
