# Peer review of "Radiomics Combined with Multiple Machine Learning Algorithms in Differentiating Pancreatic Ductal Adenocarcinoma from Pancreatic Neuroendocrine Tumor: More Hands Produce a Stronger Flame"

_jcm, 2022, doi:10.3390/jcm11226789_

Round 1
Reviewer 1 Report
Zhang et al. present an analysis of radionics based machine learning algorithm for differentiation of PDAC and NET. The Gradient Bossing Decision Tree and Random Forest model reached a sensitivity and sensitivity of 0.74 and 0.93, respectively.
It was a retrospective study including n=238 patients in a 6-year period from 2013 to 2019
Comments:
- Introduction: „Pancreatic cancer is one of the most common malignancies with an increasing inci- 40 dence and is the seventh leading cause of cancer death…“. Pancreatic cancer is the fourth leading cause of cancer-related death in the western world. Please explain the discrepancy.
- the mean age of the patients with PDAC was extremely young (59 years), whereas usually PDAC is diagnosed at a mean age of 72 years. Please again, explain the possible reasons for this difference in your cohort.
- 2.2. „All selected patients underwent CT examinations, the operating parameters of which 104 was reported in previous studies“. The parameters and contrast phases of the CT exams are important for this study and should be included in the manuscript. Were all CT scans of the 238 patients performed in the same manner at the same hospital?
- There are some typos throughout the manuscript, eg, page 7: The lack of effective diagnostic methods due to non-specific clinical symptoms may delay the diagnosis 235 and lead to larger primary tumors or metastases..
- Table 1: Please indicate: how many patients of each subgroup had available lab values for each parameter.
- Please provide the positive and negative predictive value for the GBDT and RF model as well.
- Please add the approval number of the Ethical Committee.
Author Response
Dear Reviewer,
I am very grateful for your comments for the manuscript. According with your comments and suggestions, we have revised the corresponding parts of the manuscript.
Your comments were answered below:
1)Reviewer‘s comment 1:Introduction: „Pancreatic cancer is one of the most common malignancies with an increasing incidence and is the seventh leading cause of cancer death…“. Pancreatic cancer is the fourth leading cause of cancer-related death in the western world. Please explain the discrepancy.
Answer:Thank you very much for your professional comments.
According to previous research statistics, pancreatic cancer is indeed the seventh leading cause of cancer death in men and women worldwide. Here are the references:
1. Bray F, Ferlay J, Soerjomataram I, Siegel RL, Torre LA, Jemal A (2018) Global cancer statistics 2018: GLOBOCAN estimates of incidence and mortality worldwide for 36 cancers in 185 countries. CA Cancer J Clin 68:394-424
2 Mizrahi JD, Surana R, Valle JW, Shroff RT (2020) Pancreatic cancer. Lancet 395:2008-2020
3 Sung H, Ferlay J, Siegel RL et al (2021) Global Cancer Statistics 2020: GLOBOCAN Estimates of Incidence and Mortality Worldwide for 36 Cancers in 185 Countries. CA Cancer J Clin 71:209-249
It is also true that pancreatic cancer is the fourth leading cause of cancer death in both men and women in the Western world, according to previous research statistics. Here are the reference:
Siegel RL, Miller KD, Fuchs HE, Jemal A. Cancer Statistics, 2021. CA Cancer J Clin. 2021 Jan;71(1):7-33. doi: 10.3322/caac.21654. Epub 2021 Jan 12. Erratum in: CA Cancer J Clin. 2021 Jul;71(4):359. PMID: 33433946.
One reason for this discrepancy we believe is the difference in incidence rates. The incidence of pancreatic cancer in Western countries is higher than in other countries (shown in figure below), and the possible explanation is that the diet in Western countries leads to a higher intake of meat and a higher rate of obesity (obesity happens to be an important risk factor for pancreatic cancer).Similar risk factors include diabetes and alcohol consumption. In Western countries, the high incidence of pancreatic cancer means that more patients die. This has led to pancreatic cancer becoming one of the leading causes of cancer death.
Another reason for the discrepancy, we believe, is the difference in the ability to treat cancer and the level of medical care in different regions or countries. For the Western world (e.g., the U.S.), the other cancer types that lead to cancer deaths more than pancreatic cancer include lung&bronchus cancer, breast cancer (women), prostate cancer (men), and colorectal cancer. When discussing the causes of cancer deaths globally, stomach, liver and esophageal cancers are all ranked ahead of pancreatic cancer. These GI tumors are most prevalent in Asia and Africa (especially in East Asia), while patients in Western countries often have a better prognosis due to better medical care and early detection and treatment. And improvements in diagnostic and cancer registration practices may also be in play in some countries.
In summary, other tumors (especially GI tract tumors) have higher mortality rates in other regions of the world, which makes pancreatic cancer the seventh leading cause of death from tumors worldwide, while it can rank fourth in Western countries.
Considering this very valuable suggestion, we have added this part of the description in the first paragraph of the discussion section.
2)Reviewer‘s comment 2:the mean age of the patients with PDAC was extremely young (59 years), whereas usually PDAC is diagnosed at a mean age of 72 years. Please again, explain the possible reasons for this difference in your cohort.
Answer: Thank you very much for your professional comments.
This mean age data was extracted and analyzed from real patient medical records. We believe the possible reasons are as follows:
1. Due to the development of economic level and the advancement of detection technology, the willingness and frequency of medical checkups among the population have increased, which may lead to a higher detection rate of early tumors in young patients.
2. Because our study exclusion criteria included a history of other malignancies and a history of any cancer treatment, a number of PDAC patients were excluded from our study, and we thought it was likely that these excluded patients were older patients. Our hospital is one of the top medical institutions in China, with potential patient sources covering hundreds of millions of people in several provinces of southwest China. This makes our center's medical resources very tight, and most patients are initially treated at local hospitals after the cancer was diagnosed, before a portion of patients finally come to our hospital in pursuit of better treatment. This side by side confirms the veracity of our results, as only relatively young patients, whose serve as the main economic source for their families. The willingness to come to the top regional medical institutions for treatment is strong.
3. Perhaps the age of onset of PDAC is or has been trending younger. The mean age of PDAC patients reported in many studies in recent years is below 70 years.
For example, in a study that used CT radiomics to preoperatively predict survival in PDAC patients, they reported a mean age of 67 ± 11 years for PDAC patients.(DOI: 10.2214/AJR.20.23490)
In another study that used CT radiomics and machine learning to predict tumor- stroma ratios in PDAC patients, they reported a mean age of around 60 years, similar to the results of our study. (DOI: 10.3389/fonc.2021.707288)
We have added a discussion of this issue at the end of the third paragraph of the Discussion section, based on your suggestion.
3)Reviewer‘s comment 3:2.2. „All selected patients underwent CT examinations, the operating parameters of which 104 was reported in previous studies“. The parameters and contrast phases of the CT exams are important for this study and should be included in the manuscript. Were all CT scans of the 238 patients performed in the same manner at the same hospital?
Answer: Thank you very much for your professional and constructive comments. All CT scans for these 238 patients were performed in the same manner at the same hospital. In fact we have already described the parameters of the scanning procedure including the contrast dose and the scanning delay time after injection of the media in the supplementary material. To make it more convenient for you to review, we show these details directly below:
The contrast agent we used was iohexol (300 mg iodine/mL; Bayer Schering Pharma AG, Leverkusen, Germany). The contrast agent was intravenously administered to patients prior to the contrast-enhanced CT scan to a total dose according to the dose-to-weight band (1.5 mL/kg). The contrast agent was injected at a constant rate of 2.5–3.0 mL/s using a dual‐syringe power injector (Stellant D Dual Syringe, Medrad, Indianola, PA, USA). The scanning was performed using a single 64-detector row scanner (Brilliance 64, Philips Medical Systems, Eindhoven, Netherlands). Scanning delays used in these dual-phase protocols were 30-35 seconds for early or arterial phase and 60-70 seconds for portal venous phase. Images obtained and analyzed were from the portal venous phase. The uniform scan parameters were: beam pitch, 0.891; tube voltage, 120kVp; tube current, 200mAs; detector collimation, 0.75mm; slice thickness, 1.0mm; reconstruction increment, 5.0mm; rotation time, 0.42s; and matrix, 512x512.
The descriptions were so similar to those of previous studies at the same institution that we removed them to avoid being identified by the journal's detection system as having plagiarism. In fact, we made many attempts, but since these CT scan operations are standard and fixed, no matter how much we changed the words whose meaning is similar or changed the order of the descriptions, it was ineffective. Nevertheless, we think your suggestion is valuable. Based on your suggestion, we have adjusted the description of the paragraph to avoid readers overlooking supplementary material.
4)Reviewer‘s comment 4:There are some typos throughout the manuscript, eg, page 7: The lack of effective diagnostic methods due to non-specific clinical symptoms may delay the diagnosis 235 and lead to larger primary tumors or metastases.
Answer: Thank you very much for your kind and detailed suggestion. We are sorry that these mistakes have caused you a bad review experience. We have corrected the sentence and also checked the manuscript for other language errors. During the follow-up process, if you have no other suggestions for content, but still feel that the quality of the manuscript does not meet the language requirements, we will use the journal's language editing services to make our manuscript meet the publication requirements.
5)Reviewer‘s comment 5: Table 1: Please indicate: how many patients of each subgroup had available lab values for each parameter.
Answer:Thanks for your constructive suggestions. We have revised Table 1 by adding the corresponding number of patients for each laboratory parameter.
6)Reviewer‘s comment 6: Please provide the positive and negative predictive value for the GBDT and RF model as well.
Answer: Thank you very much for your suggestion.
In the training dataset, the positive and negative prediction values were 0.950 and 0.886, respectively; in the validation dataset, the positive and negative prediction values were 0.898 and 0.822, respectively. We have added these results to the corresponding paragraphs in the Results 3.3 section. Thanks again for your helpful suggestions.
7)Reviewer‘s comment 7:Please add the approval number of the Ethical Committee.
Answer: Thank you very much for your kind and detailed suggestion. In fact we have stated in the "Institutional Review Board Statement" section that we have received approval from the Institutional Ethics Committee under the approval number 2019-903. However, based on your suggestion, we have made a small revision in that section to make it clear to the reader.
Finally we thank you for your professional and helpful comments and suggestions.
Yours sincerely,
Hao Zhang
Reviewer 2 Report
Authors evaluated the diagnostic ability of radiomics features combined with multiple machine learning algorithms in differentiating PDCA from p-NET, and concluded that based on enhanced CT radiomics features combined with multiple machine learning algorithms have a promising ability to differentiate PDCA from p-NET.
However, the method of 'enhanced CT' or the diagnostic criteria of PDCA and p-NET were not shown. These were key points of the diagnosis of diagnostic imaging of pancreatic tumors.
Therefore authors were recommend to describe more in detail about the method of the CT examination as below.
1. Authors were recommended to describe the amount of the contrast media and the scanning delay time after injection of the media.
2. Authors were recommended to describe the diagnostic criteria of PDAC from p-NET using ROI data of the enhanced CT.
Author Response
Dear Reviewer,
I am very grateful for your comments for the manuscript. According with your comments and suggestions, we have revised the corresponding parts of the manuscript.
Some of your questions were answered below:
1)Reviewer‘s comment 1:Authors were recommended to describe the amount of the contrast media and the scanning delay time after injection of the media.
Answer: Thank you very much for your professional and constructive comments. In fact we have described the parameters of the scanning procedure including the contrast dose and the scanning delay time after injection of the media in the supplementary material. To make it more convenient for you to review, we show these details directly below:
The contrast agent we used was iohexol (300 mg iodine/mL; Bayer Schering Pharma AG, Leverkusen, Germany). The contrast agent was intravenously administered to patients prior to the contrast-enhanced CT scan to a total dose according to the dose-to-weight band (1.5 mL/kg). The contrast agent was injected at a constant rate of 2.5–3.0 mL/s using a dual‐syringe power injector (Stellant D Dual Syringe, Medrad, Indianola, PA, USA). The scanning was performed using a single 64-detector row scanner (Brilliance 64, Philips Medical Systems, Eindhoven, Netherlands). Scanning delays used in these dual-phase protocols were 30-35 seconds for early or arterial phase and 60-70 seconds for portal venous phase. Images obtained and analyzed were from the portal venous phase. The uniform scan parameters were: beam pitch, 0.891; tube voltage, 120kVp; tube current, 200mAs; detector collimation, 0.75mm; slice thickness, 1.0mm; reconstruction increment, 5.0mm; rotation time, 0.42s; and matrix, 512x512.
The descriptions were so similar to those of previous studies at the same institution that we removed them to avoid being identified by the journal's detection system as having plagiarism. In fact, we made many attempts, but since these CT scan operations are standard and fixed, no matter how much we changed the words whose meaning is similar or changed the order of the descriptions, it was ineffective. Nevertheless, we think your suggestion is valuable. Based on your suggestion, we have adjusted the description of the paragraph to avoid readers overlooking supplementary material.
2)Reviewer‘s comment 2:Authors were recommended to describe the diagnostic criteria of PDAC from p-NET using ROI data of the enhanced CT.
Answer: Thank you very much for your professional and constructive suggestion. In section 2.2, we have added a description of the diagnostic criteria of enhanced CT for pancreatic tumors.
In addition to the above revisions, we have made the following revisions:
1. We have added a discussion of regional differences in mortality from pancreatic cancer at the beginning of the first paragraph of the discussion section.
2. We have also added a discussion of the differences in the mean age of onset of PDAC at the end of the third paragraph of the Discussion section.
3. We have corrected the sentence and also checked the manuscript for other language errors. During the follow-up process, if you have no other suggestions for content, but still feel that the quality of the manuscript does not meet the language requirements, we will use the journal's language editing services to make our manuscript meet the publication requirements.
4. We revised Table 1 in order to clarify how many patients in each subgroup had available laboratory parameters.
5. In the corresponding paragraph of the Results 3.3 section, we have added a description of the positive predictive values and negative predictive values.
6. We have made a small revision in the “Institutional Review Board Statement” section to clarify the ethical approval number for the reader's understanding.
Finally we thank you for your professional and helpful comments and suggestions.
Yours sincerely,
Hao Zhang